# MULTIPLICATIVE LSTM FOR SEQUENCE MODELLING

**Ben Krause, Iain Murray & Steve Renals**
School of Informatics, University of Edinburgh
Edinburgh, Scotland, UK
{ben.krause,i.murray,s.renals}@ed.ac.uk

**Liang Lu**
Toyota Technological Institute at Chicago
Chicago, Illinois, USA
{llu}@ttic.edu

## ABSTRACT

We introduce multiplicative LSTM (mLSTM), a novel recurrent neural network architecture for sequence modelling that combines the long short-term memory (LSTM) and multiplicative recurrent neural network architectures. mLSTM is characterised by its ability to have different recurrent transition functions for each possible input, which we argue makes it more expressive for autoregressive density estimation. We demonstrate empirically that mLSTM outperforms standard LSTM and its deep variants for a range of character level modelling tasks, and that this improvement increases with the complexity of the task. This model achieves a test error of 1.19 bits/character on the last 4 million characters of the Hutter prize dataset when combined with dynamic evaluation.

## 1 INTRODUCTION

Recurrent neural networks (RNNs) are powerful sequence density estimators that can use long contexts to make predictions. They have achieved tremendous success in (conditional) sequence modelling tasks such as language modelling, machine translation and speech recognition. Generative models of sequences can apply factorization via the product rule to perform density estimation of the sequence $x_{1:T} = \{x_1, \ldots, x_T\}$,

$$P(x_1, \ldots, x_T) = P(x_1)P(x_2|x_1)P(x_3|x_2, x_1) \cdots P(x_T|x_1 \ldots x_{T-1}). \tag{1}$$

RNNs can model sequences with the above factorization by using a hidden state to summarize past inputs. The hidden state vector $h_t$ is updated recursively using the previous hidden state vector $h_{t-1}$ and the current input $x_t$ as

$$h_t = \mathcal{F}(h_{t-1}, x_t), \tag{2}$$

where $\mathcal{F}$ is a differentiable function with learnable parameters. In a vanilla RNN, $\mathcal{F}$ multiplies its inputs by a matrix and squashes the result with a non-linear function such as a hyperbolic tangent (tanh). The updated hidden state vector is then used to predict a probability distribution over the next sequence element, using function $\mathcal{G}$. In the case where $x_{1:T}$ consists of mutually exclusive discrete outcomes, $\mathcal{G}$ may apply a matrix multiplication followed by a softmax function:

$$P(x_{t+1}) = \mathcal{G}(h_t). \tag{3}$$

Generative RNNs can evaluate log-likelihoods of sequences exactly, and are differentiable with respect to these log-likelihoods. RNNs can be difficult to train due to the vanishing gradient problem (Bengio et al., 1994), but advances such as the long short-term memory architecture (LSTM) (Hochreiter & Schmidhuber, 1997) have allowed RNNs to be successful. Despite their success, generative RNNs (as well as other conditional generative models) are known to have problems with recovering from mistakes (Graves, 2013). Each time the recursive function of the RNN is applied and the hidden state is updated, the RNN must decide which information from the previous hidden state to store, due to its limited capacity. If the RNN's hidden representation remembers the wrong information and reaches a bad numerical state for predicting future sequence elements, for instance as a result of an unexpected input, it may take many time-steps to recover.

We argue that RNN architectures with hidden-to-hidden transition functions that are input-dependent are better suited to recover from surprising inputs. Our approach to generative RNNs combines LSTM units with multiplicative RNN (mRNN) factorized hidden weights, allowing flexible input-dependent

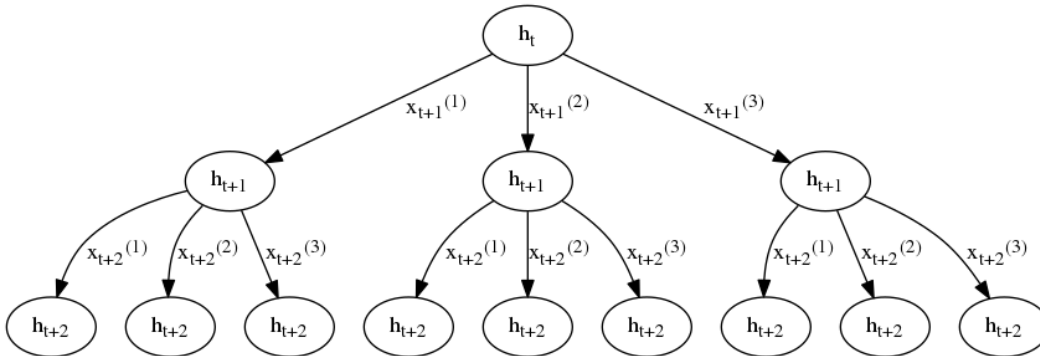

Figure 1: Diagram of hidden states of a generative RNN as a tree, where $x_t^{(n)}$ denotes which of $N$ possible inputs is encountered at timestep $t$. Given $h_t$, the starting node of the tree, there will be a different possible $h_{t+1}$ for every $x_{t+1}^{(n)}$. Similarly, for every $h_{t+1}$ that can be reached from $h_t$, there is a different possible $h_{t+2}$ for each $x_{t+2}^{(n)}$, and so on.

transitions that are easier to control due to the gating units of LSTM . We compare this multiplicative LSTM hybrid architecture with other variants of LSTM on a range of character level language modelling tasks. Multiplicative LSTM is most appropriate when it can learn parameters specifically for each possible input at a given timestep. Therefore, its main application is to sequences of discrete mutually exclusive elements, such as language modelling and related problems.

## 1.1 INPUT-DEPENDENT TRANSITION FUNCTIONS

RNNs learn a mapping from previous hidden state $h_{t-1}$ and input $x_t$ to hidden state $h_t$. Let $\hat{h}_t$ denote the input to the next hidden state before any non-linear operation:

$$\hat{h}(t) = W_{hh}h_{t-1} + W_{hx}x_t, \tag{4}$$

where $W_{hh}$ is the hidden-to-hidden weight matrix, and $W_{hx}$ is the input-to-hidden weight matrix. For problems such as language modelling, $x_t$ is a one-hot vector, meaning that the output of $W_{hx}x_t$ is a column in $W_{hx}$, corresponding to the unit element in $x_t$.

The possible future hidden states in an RNN can be viewed as a tree structure, as shown in Figure 1. For an alphabet of $N$ inputs and a fixed $h_{t-1}$, there will be $N$ possible transition functions between $h_{t-1}$ and $\hat{h}_t$. The relative magnitude of $W_{hh}h_{t-1}$ to $W_{hx}x_t$ will need to be large for the RNN to be able to use long range dependencies, and the resulting possible hidden state vectors will therefore be highly correlated across the possible inputs, limiting the width of the tree and making it harder for the RNN to form distinct hidden representations for different sequences of inputs. However, if the RNN has flexible input-dependent transition functions, the tree will be able to grow wider more quickly, giving the RNN the flexibility to represent more probability distributions.

In a vanilla RNN, it is difficult to allow inputs to greatly affect the hidden state vector without erasing information from the past hidden state. However, an RNN with a transition function mapping $\hat{h}_t \leftarrow h_{t-1}$ dependent on the input would allow the relative values of $h_t$ to vary with each possible input $x_t$, without overwriting the contribution from the previous hidden state, allowing for more long term information to be stored. This ability to adjust to new inputs quickly while limiting the overwriting of information should make an RNN more robust to mistakes when it encounters surprising inputs, as the hidden vector is less likely to get trapped in a bad numerical state for making future predictions.

## 1.2 MULTIPLICATIVE RNN

The multiplicative RNN (mRNN) (Sutskever et al., 2011) is an architecture designed specifically to allow flexible input-dependent transitions. Its formulation was inspired by the tensor RNN, an RNN architecture that allows for a different transition matrix for each possible input. The tensor RNN

features a 3-way tensor $W_{hh}^{1:N}$, which contains a separately learned transition matrix $W_{hh}$ for each input dimension. The 3-way tensor can be stored as an array of matrices

$$W_{hh}^{(1:N)} = \{W_{hh}^{(1)}, ..., W_{hh}^{(N)}\}, \tag{5}$$

where superscript is used to denote the index in the array, and $N$ is the dimensionality of $x_t$. The specific hidden-to-hidden weight matrix $W_{hh}^{(x_t)}$ used for a given input $x_t$ is then

$$W_{hh}^{(x_t)} = \sum_{n=1}^{N} W_{hh}^{(n)} x_t^{(n)}. \tag{6}$$

For language modelling problems, only one unit of $x_t$ will be on, and $W_{hh}^{(x_t)}$ will be the matrix in $W_{hh}^{(1:N)}$ corresponding to that unit. Hidden-to-hidden propagation in the tensor RNN is then given by

$$\hat{h}(t) = W_{hh}^{(x_t)} h_{t-1} + W_{hx} x_t. \tag{7}$$

The large number of parameters in the tensor RNN make it impractical for most problems. mRNNs can be thought of as a tied-parameter approximation to the tensor RNN that use a factorized hidden-to-hidden transition matrix in place of the normal RNN hidden-to-hidden matrix $W_{hh}$, with an input-dependent intermediate diagonal matrix $\mathrm{diag}(W_{mx} x_t)$. The input-dependent hidden-to-hidden weight matrix, $W_{hh}^{(x_t)}$ is then

$$W_{hh}^{(x_t)} = W_{hm} \mathrm{diag}(W_{mx} x_t) W_{mh}. \tag{8}$$

An mRNN is thus equivalent to a tensor RNN using the above form for $W_{hh}^{(x_t)}$. For readability, an mRNN can also be described using intermediate state $m_t$ as follows:

$$m_t = (W_{mx} x_t) \odot (W_{mh} h_{t-1}) \tag{9}$$

$$\hat{h}_t = W_{hm} m_t + W_{hx} x_t. \tag{10}$$

mRNNs have improved on vanilla RNNs at character level language modelling tasks (Sutskever et al., 2011; Mikolov et al., 2012), but have fallen short of the more popular LSTM architecture, for instance as shown with LSTM baselines from (Cooijmans et al., 2016). The standard RNN units in an mRNN do not provide an easy way for information to bypass its complex transitions, resulting in the potential for difficulty in retaining long term information.

## 1.3 Long short-term memory

LSTM is a commonly used RNN architecture that uses a series of multiplicative gates to control how information flows in and out of internal states of the network (Hochreiter & Schmidhuber, 1997). There are several slightly different variants of LSTM, and we present the variant used in our experiments.

The LSTM hidden state receives inputs from the input layer $x_t$ and the previous hidden state $h_{t-1}$:

$$\hat{h}_t = W_{hx} x_t + W_{hh} h_{t-1}. \tag{11}$$

The LSTM network also has 3 gating units – input gate $i$, output gate $o$, and forget gate $f$ – that have both recurrent and feed-forward connections:

$$i_t = \sigma(W_{ix} x_t + W_{ih} h_{t-1}) \tag{12}$$

$$o_t = \sigma(W_{ox} x_t + W_{oh} h_{t-1}) \tag{13}$$

$$f_t = \sigma(W_{fx} x_t + W_{fh} h_{t-1}), \tag{14}$$

where $\sigma$ is the logistic sigmoid function. The input gate controls how much of the input to each hidden unit is written to the internal state vector $c_t$, and the forget gate determines how much of the previous internal state $c_{t-1}$ is preserved. This combination of write and forget gates allows the network to control what information should be stored and overwritten across each time-step. The internal state is updated by

$$c_t = f_t \odot c_{t-1} + i_t \odot \hat{h}_t. \tag{15}$$

The output gate controls how much of each unit's activation is preserved. It allows the LSTM cell to keep information that is not relevant to the current output, but may be relevant later. The final output of the hidden state is given by

$$h_t = \tanh(c_t \odot o_t). \tag{16}$$

This is slightly different from the typical LSTM variant, where the output gate is applied after the $\tanh$. LSTM's ability to control how information is stored in each unit has proven generally useful.

### 1.4 COMPARING LSTM WITH MRNN

The LSTM and mRNN architectures both feature multiplicative units, but these units serve different purposes. LSTM's gates are designed to control the flow of information through the network, whereas mRNN's gates are designed to allow transition functions to vary across inputs. LSTM gates receive input from both the input units and hidden units, allowing multiplicative interactions between hidden units, but also potentially limiting the extent of input-hidden multiplicative interaction. LSTM gates are also squashed with a sigmoid, forcing them to take values between 0 and 1, which makes them easier to control, but less expressive than mRNN's linear gates. For language modelling problems, mRNN's linear gates do not need to be controlled by the network because they are explicitly learned for each input. They are also placed in between a product of 2 dense matrices, giving more flexibility to the possible values of the final product of matrices.

## 2 MULTIPLICATIVE LSTM

Since the LSTM and mRNN architectures are complimentary, we propose the multiplicative LSTM (mLSTM), a hybrid architecture that combines the factorized hidden-to-hidden transition of mRNNs with the gating framework from LSTMs. The mRNN and LSTM architectures can be combined by adding connections from the mRNN's intermediate state $m_t$ (which is redefined below for convenience) to each gating units in the LSTM, resulting in the following system:

$$m_t = (W_{mx}x_t) \odot (W_{mh}h_{t-1}) \tag{17}$$

$$\hat{h}_t = W_{hx}x_t + W_{hm}m_t \tag{18}$$

$$i_t = \sigma(W_{ix}x_t + W_{im}m_t) \tag{19}$$

$$o_t = \sigma(W_{ox}x_t + W_{om}m_t) \tag{20}$$

$$f_t = \sigma(W_{fx}x_t + W_{fm}m_t). \tag{21}$$

We set the dimensionality of $m_t$ and $h_t$ equal for all our experiments. We also chose to share $m_t$ across all LSTM unit types, resulting in a model with 1.25 times the number of recurrent weights as LSTM for the same number of hidden units.

The goal of this architecture is to combine the flexible input-dependent transitions of mRNNs with the long time lag and information control of LSTMs. The gated units of LSTMs could make it easier to control (or bypass) the complex transitions in that result from the factorized hidden weight matrix. The additional sigmoid input and forget gates featured in LSTM units allow even more flexible input-dependent transition functions than in regular mRNNs.

## 3 RELATED APPROACHES

Many recently proposed RNN architectures use recurrent depth, which is depth between recurrent steps. Recurrent depth allows more non-linearity in the combination of inputs and previous hidden states from every time step, which in turn allows for more flexible input-dependent transitions. Recurrent depth has been found to perform better than other kinds of non-recurrent depth for sequence modelling (Zhang et al., 2016). Recurrent highway networks (Zilly et al., 2016) use a more sophisticated recurrent depth that carefully controls propagation through layers using gating units. The gating units also allow for a greater deal of multiplicative interaction between the inputs and hidden units. While adding recurrent depth could improve our model, we believe that maximizing the input-dependent flexibility of the transition function is more important for expressive sequence modelling. Recurrent depth can do this through non-linear layers combining hidden and input contributions, but our method can do this independently of depth.

| architecture | test set error (bits/char) |
|---|---|
| mRNN (Mikolov et al., 2012) | 1.41 |
| multiplicative integration RNN (Wu et al., 2016) | 1.39 |
| LSTM (Cooijmans et al., 2016) | 1.38 |
| **mLSTM** | **1.35** |
| batch normalized LSTM (Cooijmans et al., 2016) | 1.32 |
| zoneout RNN (Krueger et al., 2016) | 1.30 |
| hierarchical multiscale LSTM (Chung et al., 2016) | 1.27 |
| 2-layer norm hyperLSTM (Ha et al., 2016) | 1.22 |

Table 1: Test set error (bits/char) on Penn Treebank dataset for mLSTM compared with past work.

Another approach, multiplicative integration RNNs (Wu et al., 2016), use Hadamard products instead of addition when combining contributions from input and hidden units. When applying this to LSTM, this architecture achieves impressive sequence modelling results. The main difference between multiplicative integration LSTM and mLSTM is that mLSTM applies the Hadamard product between the multiplication of two matrices. In the case of LSTM, this allows for the potential for greater expressiveness, without significantly increasing the size of the model.

# 4 EXPERIMENTS

## 4.1 SYSTEM SETUP

Our experiments compared the performance of mLSTM with regular LSTM for different character-level language modelling tasks of varying complexity[1]. Gradient computation in these experiments used truncated backpropagation through time on sequences of length 100, only resetting the hidden state every 10 000 timesteps to allow networks access to information far in the past. All experiments used a variant of RMSprop, (Tieleman & Hinton, 2012), with normalized updates in place of a learning rate. All unnormalized update directions $v_*$, computed by RMSprop, were normalized to have length $\ell$, where $\ell$ was decayed exponentially over training:

$$v \leftarrow \frac{\ell}{\sqrt{v_*^T v_*}} v_*. \tag{22}$$

We found that this allowed for fast convergence with larger batch sizes, allowing for greater parallelization during training without hurting performance.

We compared mLSTM to previously reported regular LSTM, stacked LSTM, and RNN character-level language models. The stacked LSTMs were all 2-layer, and both LSTM layers contained direct connections from the inputs and to the outputs. We used the Penn Treebank dataset (Marcus et al., 1993) to test small scale language modelling, the processed and raw versions of the Wikipedia text8 dataset (Hutter, 2012) to test large scale language modelling and byte level language modelling respectively, and the European parliament dataset (Koehn, 2005) to investigate multilingual fitting.

## 4.2 PENN TREEBANK

The Penn treebank dataset is relatively small, and consists of only case insensitive English characters, with no punctuation. It is one of the most widely used language modelling bench mark tasks. Due to its small size, the main bottleneck for performance is overfitting.

We fitted an mLSTM with 700 hidden units to the Penn Treebank dataset, with no regularization other than early stopping. We used a slightly different version of this dataset, where the frequently occurring token <unk> was replaced by a single character, shortening the file by about 4%. To make our results comparable to other results on this dataset, we computed the total cross entropy of the test set file and divided this by the number of characters in the original test set. The results are shown in

---

[1] Code to replicative our large scale experiments on the Hutter prize dataset is available at `https://github.com/benkrause/mLSTM`.

| architecture | test set error (bits/char) |
|---|---|
| small LSTM (Zhang et al., 2016) | 1.65 |
| **small LSTM (ours)** | **1.64** |
| small deep LSTM (best) (Zhang et al., 2016) | 1.63 |
| **small mLSTM** | **1.59** |
| mRNN (Mikolov et al., 2012) | 1.54 |
| multiplicative integration RNN (Wu et al., 2016) | 1.52 |
| skipping RNN (Pachitariu & Sahani, 2013) | 1.48 |
| multiplicative integration LSTM (Wu et al., 2016) | 1.44 |
| LSTM (Cooijmans et al., 2016) | 1.43 |
| **mLSTM** | **1.40** |
| batch normalised LSTM (Cooijmans et al., 2016) | 1.36 |
| hierarchical multiscale LSTM (Chung et al., 2016) | 1.30 |

Table 2: Text8 dataset test set error in bits/char. Architectures labelled with small used a highly restrictive hidden dimensionality (512 for LSTM, 450 for LSTM)

Table 1, where it can be seen that mLSTM achieved 1.35 bits/char test set error, compared with 1.38 bits/char for an unregularized LSTM (Cooijmans et al., 2016).

## 4.3 TEXT8 DATASET

Text8 contains 100 million characters of English text taken from Wikipedia in 2006, consisting of just the 26 characters of the English alphabet plus spaces. This dataset can be found at `http://mattmahoney.net/dc/textdata`. This corpus has been widely used to benchmark RNN character level language models, with the first 90 million characters used for training, the next 5 million used for validation, and the final 5 million used for testing. The results of these experiments are shown in Table 2.

The first set of experiments we performed were designed to be comparable to those of Zhang et al. (2016), who benchmarked several deep LSTMs against shallow LSTMs on this dataset. The shallow LSTM had a hidden state dimensionality of 512, and the deep versions had reduced dimensionality to give them roughly the same number of parameters. Our experiment used an mLSTM with a hidden dimensionality of 450, giving it slightly fewer parameters than the past work, and our own LSTM baseline with hidden dimensionality 512. mLSTM showed an improvement over our baseline and the previously reported best deep LSTM variant.

We ran additional experiments to compare a large mLSTM with other reported experiments. We trained an mLSTM with hidden dimensionality of 1900 on the text8 dataset. mLSTM was able to fit the training data well and achieved a competitive performance; however it was outperformed by other architectures that are less prone to over-fitting.

## 4.4 HUTTER PRIZE DATASET

We performed experiments using the raw version of the Wikipedia dataset, originally used for the Hutter Prize compression benchmark (Hutter, 2012). This dataset consists mostly of English language text and mark-up language text, but also contains text in other languages, including non-Latin languages. The dataset is modelled using a UTF-8 encoding, and contains 205 unique bytes.

We compared mLSTMs and 2-layer stacked LSTMs for varying network sizes, ranging from about 3–20 million parameters. These results all used RMS prop with normalized updates, stopping after 4 epochs, with test performance measured on the last 5 million bytes. Hyperparameters for each mLSTM and stacked LSTM were kept constant across all sizes, and were reused from the previous experiment using the text8 dataset. The results, shown in Figure 2, show that mLSTM gives a modest improvement across all network sizes.

We hypothesized that mLSTM's superior performance over stacked LSTM was in part due to its ability to recover from surprising inputs. To test this we looked at each network's performance after viewing surprising inputs that occurred naturally in the test set by creating a set of the 10% characters

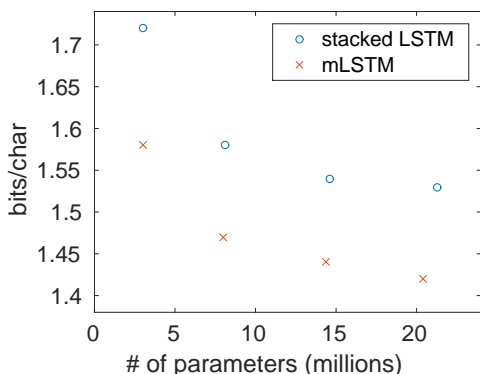

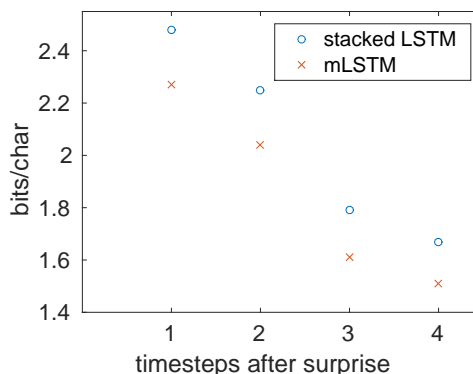

Figure 2: Hutter prize validation performance in bits/char plotted against number of network parameters for mLSTM and stacked LSTM.

Figure 3: Cross entropy loss for mLSTM and stacked LSTM immediately proceeding a surprising input

with the largest average loss taken by mLSTM and stacked LSTM. Both networks perform roughly equally on this set of surprising characters, with mLSTM and stacked LSTM taking losses of 6.27 bits/character and 6.29 bits/character respectively. However, stacked LSTM tended to take much larger losses than mLSTM in the timesteps immediately following surprising inputs. One to four time-steps after a surprising input occurred, mLSTM and stacked LSTM took average losses of (2.26, 2.04, 1.61, 1.51) and (2.48, 2.25, 1.79, 1.67) bits per character respectively, as shown in Figure 3. mLSTM's overall advantage over stacked LSTM was 1.42 bits/char to 1.53 bits/char; mLSTM's advantage over stacked LSTM was greater after a surprising input than it is in general.

We also tested our largest mLSTM and stacked LSTM models using dynamic evaluation, where the network's weights are adapted to fit recent sequences (Graves, 2013). Dynamic evaluation can be thought of as a type of fast weights memory structure (Ba et al., 2016) that draws the network towards regularities that recently occurred in the sequence, causing the network to assign higher probabilities to these regularities occurring again soon. Unlike other approaches to fast weights where the network learns to control the weights, dynamic evaluation uses the error signal and gradients to update the weights, which potentially increases its effectiveness, but also limits its scope to conditional generative modelling, when the outputs can be observed after they are predicted. Rather than performing fully online dynamic evaluation, we adapted incrementally to short sequences, allowing for gradients to be passed back over longer time scales. Thus, we divided the test set into sequences of length 50 in order of occurrence. After predicting a sequence and incurring a loss, we trained the RNN for a single iteration on that sequence, using RMSprop (with a learning rate, as normalized updates no longer make sense in a stochastic setting), and weight decay. After updating the RNN, we recomputed the forward pass through this sequence to update the final hidden state. The updated RNN was then used to predict the next sequence of 50 elements, and this process was repeated. Dynamic evaluation was applied to a test set of the last 4 million bytes (instead of last 5 million) to be comparable with the only other dynamic evaluation result (Graves, 2013).

The best results for stacked LSTM and mLSTM are given in Table 3, alongside results from the literature. mLSTM performs at near state-of-the-art level when evaluated statically, and greatly outperforms the best static models and other dynamic models when evaluated dynamically.

## 4.5 MULTILINGUAL LEARNING

Character-level RNNs are able to simultaneously model multiple languages using shared parameters. These experiments compared the relative ability of LSTM and mLSTM at fitting a dataset in a single language and a combined dataset with two separate languages. We used the first 100 million characters of the English and Spanish translations of the European parliament dataset to make an English dataset and a Spanish dataset. Each dataset was split 90-5-5 for training, validation, and testing. We created a third Spanish-English hybrid dataset by combining the Spanish and English datasets, resulting in a dataset twice as large. These datasets were left in their raw form, containing punctuation and both

| architecture | test error (bits/char) |
|---|---|
| 7-layer stacked LSTM (Graves, 2013) | 1.67 |
| gf-LSTM (Chung et al., 2015) | 1.58 |
| **2-layer stacked LSTM (ours)** | **1.53** |
| grid LSTM (Kalchbrenner et al., 2015) | 1.48 |
| multiplicative integration LSTM (Wu et al., 2016) | 1.44 |
| **mLSTM** | **1.42** |
| hierarchical multiscale RNNs (Chung et al., 2016) | 1.40 |
| recurrent memory array structures (Rocki, 2016a) | 1.40 |
| feedback LSTM (Rocki, 2016b) | 1.39* |
| feedback LSTM + zoneout (Rocki, 2016b) | 1.37* |
| layer norm hyperLSTM (Ha et al., 2016) | 1.34 |
| 7-layer stacked LSTM (dynamic) (Graves, 2013) | 1.33* |
| bytenet decoder (Kalchbrenner et al., 2016) | 1.33 |
| **2-layer stacked LSTM (ours, dynamic)** | **1.32*** |
| recurrent highway networks (Zilly et al., 2016) | 1.32 |
| **mLSTM (dynamic)** | **1.19*** |

Table 3: Raw wikipedia dataset validation error in bits/char. Results labelled with * use the error signal to update the hidden state, and architectures labelled with (dynamic) use gradient descent based fitting to recent sequences to perform this adjustment.

| architecture | English only test error | Spanish only test error | Spanish-English test error |
|---|---|---|---|
| LSTM | 1.13 | 1.01 | 1.14 |
| mLSTM | 1.05 | 0.95 | 1.04 |

Table 4: European parliament test error in bits/char for LSTM and mLSTM on English only, Spanish only, and mixed English-Spanish.

upper-case and lower-case letters. The LSTMs in these experiments had a hidden dimensionality of 2200, and the mLSTMs had a hidden dimensionality of 1900. All experiments were run for 4 epochs.

mLSTM generally outperformed LSTM at this task, as shown in Table 4. However, there also seemed to be an interaction effect with the number of languages. Increasing the complexity of the task by forcing the RNN to learn 2 languages instead of 1 presented a larger fitting difficulty for LSTM than for mLSTM.

## 5 DISCUSSION

This work combined the mRNN's factorized hidden weights with the LSTM's hidden units for generative modelling of discrete multinomial sequences. This mLSTM architecture was motivated by its ability to have both controlled and flexible input-dependent transitions, to allow for fast changes to the distributed hidden representation without erasing information. In a series of character-level language modelling experiments, mLSTM showed improvements over LSTM and its deep variants. This relative improvement increased with the complexity of the task, and provides evidence that mLSTM has a more powerful fitting ability for character-level language modelling than regular LSTM and its common deep variants. mLSTM performed competitively at large scale character-level language modelling, and achieved dramatic improvement over the state of the art with 1.19 bits/character on the Hutter prize dataset when combined with dynamic evaluation, motivating further investigation of dynamic evaluation for RNN sequence modelling.

While these results are promising, it remains to be seen how mLSTM performs at word-level language modelling and other discrete multinomial generative modelling tasks, and whether mLSTM can be formulated to apply more broadly to tasks with continuous or non-sparse input units. We also hope this work will motivate further exploration in generative RNN architectures with flexible input-dependent transition functions.

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
