# Peer review of "Multiplicative LSTM for sequence modelling"

_ICLR 2017 — rejected_

[Official Review · AnonReviewer3 · rating 6 · confidence 4 · 16 Dec 2016]
**useful new LSTM variant**

Pros:
* Clearly written.
* New model mLSTM which seems to be useful according to the results.
* Some interesting experiments on big data.

Cons:
* Number of parameters in comparisons of different models is missing.
* mLSTM is behind some other models in most tasks.

[Official Review · AnonReviewer2 · rating 4 · confidence 5 · 17 Dec 2016]
**Not bad work**

This paper proposes an extension of the multiplicative RNN [1] where the authors apply the same reparametrization trick to the weight matrices of the LSTM. 

The paper proposes some interesting tricks, but none of them seems to be very crucial. For instance, in Eq. (16), the authors propose to multiply the output gate inside the activation function in order to alleviate the saturation problem in logistic sigmoid or hyperbolic tangent. Also, the authors share m_t across the inference of different gating units and cell-state candidates, at the end this brings only 1.25 times increase on the number of model parameters. Lastly, the authors use a variant of RMSProp where they add an additional hyper-parameter $\ell$ and schedule it across the training time. It would be nicer to apply the same tricks to other baseline models and show the improvement with regard to each trick.

With the new architectural modification to the LSTM and all the tricks combined, the performance is not as great as we would expect. Why didn’t the authors apply batch normalization, layer normalization or zoneout to their models? Was there any issue with applying one of those regularization or optimization techniques? 

At the fourth paragraph of Section 4.4, where the authors connect dynamic evaluation with fast weights is misleading. I find it a bit hard to connect dynamic evaluation as a variant of fast weights. Fast weights do not use test error signal. In the paper, the authors claim that “dynamic evaluation uses the error signal and gradients to update the weights, which potentially increases its effectiveness, but also limits its scope to conditional generative modelling, when the outputs can be observed after they are predicted”, and I am afraid to tell that this assumption is very misleading. We should never assume that test label information is given at the inference time. The test label information is there to evaluate the generalization performance of the model. In some applications, we may get the label information at test time, e.g., stock prediction, weather forecasting, however, in many other applications, we don’t. For instance, in machine translation, we don't know what's the best translation at the end, unlike weather forecasting. Also, it would be fair to apply dynamic evaluation to all the other baseline models as well to compare with the BPC score 1.19 achieved by the proposed mLSTM.

The quality of the work is not that bad, but the novelty of the paper is not that good either. The performance of the proposed model is oftentime worse than other methods, and it is only better when dynamic evaluation is coupled together. However, dynamic evaluation can improve the other methods as well.

[1] Ilya et al., “Generating Text with Recurrent Neural Networks”, ICML’11

[Official Review · AnonReviewer1 · rating 4 · confidence 4 · 20 Dec 2016]
**An Interesting paper**

* Brief Summary: 

This paper explores an extension of multiplicative RNNs to the LSTM type of models. The resulting proposal is very similar to [1]. Authors show experimental results on character-level language modeling tasks. In general, I think the paper is well-written and the explanations are quite clear.

* Criticisms:

- In terms of contributions, the paper is weak. The motivation makes sense, however, very similar work has been done in [1] and already an extension over [2]. Because of that this paper mainly stands as an application paper.
- The results are encouraging. On the other hand, they are still behind the state of art without using dynamic evaluation. 
- There are some non-standard choices on modifications on the standard algorithms, such as "l" parameter of RMSProp and multiplying output gate before the nonlinearity.
- The experimental results are only limited to character-level language modeling only. 

* An Overview of the Review:

Pros:
- A simple modification that seems to reasonably well in practice.
- Well-written.

Cons:
- Lack of good enough experimental results.
- Not enough contributions (almost trivial extension over existing algorithms).
- Non-standard modifications over the existing algorithms.

[1] Wu Y, Zhang S, Zhang Y, Bengio Y, Salakhutdinov RR. On multiplicative integration with recurrent neural networks. InAdvances in Neural Information Processing Systems 2016 (pp. 2856-2864).
[2] Sutskever I, Martens J, Hinton GE. Generating text with recurrent neural networks. InProceedings of the 28th International Conference on Machine Learning (ICML-11) 2011 (pp. 1017-1024).

[Final Decision · Program Chairs · 06 Feb 2017]
**ICLR committee final decision**

The paper presents a new way of doing multiplicative / tensored recurrent weights in RNNs. The multiplicative weights are input dependent. Results are presented on language modeling (PTB and Hutter). We found the paper to be clearly written, and the idea well motivated. However, as pointed out by the reviewers, the results were not state of the art. We feel that is that this is because the authors did not make a strong attempt at regularizing the training. Better results on a larger set of tasks would have probably made this paper easier to accept. 
 
 Pros:
 - interesting idea, and reasonable results
 Cons:
 - only shown on language modeling tasks
 - results were not very strong, when compared to other methods (which typically used strong regularization and training like batch normalization etc).
 - reviewers did not find the experiments convincing enough, and felt that a fair comparison would be to compare with dynamic weights on the competing RNNs.